# Spontaneous Emergence of Multicellular Heritability

**DOI:** 10.3390/genes14081635

**Published:** 2023-08-17

**Authors:** Seyed Alireza Zamani-Dahaj, Anthony Burnetti, Thomas C. Day, Peter J. Yunker, William C. Ratcliff, Matthew D. Herron

**Affiliations:** 1Interdisciplinary Graduate Program in Quantitative Biosciences, Georgia Institute of Technology, Atlanta, GA 30332, USA; dahaj1367@gmail.com; 2Georgia Institute of Technology, School of Physics, Atlanta, GA 30332, USA; tday31@gatech.edu (T.C.D.); peter.yunker@gatech.edu (P.J.Y.); 3Georgia Institute of Technology, School of Biological Sciences, Atlanta, GA 30332, USA; anthony.burnetti@biosci.gatech.edu (A.B.); xprinceps@gmail.com (M.D.H.)

**Keywords:** major evolutionary transitions, heritability, evolvability, MLS2, Darwinian individuality, evolutionary transitions in individuality

## Abstract

The major transitions in evolution include events and processes that result in the emergence of new levels of biological individuality. For collectives to undergo Darwinian evolution, their traits must be heritable, but the emergence of higher-level heritability is poorly understood and has long been considered a stumbling block for nascent evolutionary transitions. Using analytical models, synthetic biology, and biologically-informed simulations, we explored the emergence of trait heritability during the evolution of multicellularity. Prior work on the evolution of multicellularity has asserted that substantial collective-level trait heritability either emerges only late in the transition or requires some evolutionary change subsequent to the formation of clonal multicellular groups. In a prior analytical model, we showed that collective-level heritability not only exists but is usually more heritable than the underlying cell-level trait upon which it is based, as soon as multicellular groups form. Here, we show that key assumptions and predictions of that model are borne out in a real engineered biological system, with important implications for the emergence of collective-level heritability.

## 1. Introduction

Life on Earth is, and always has been, predominantly unicellular. At various times, unicellular populations have evolved multicellular structures, bodies, or thalli. In some cases, these now multicellular populations massively diversified, giving rise to the major radiations of multicellular life: plants, animals, fungi, and various groups of seaweeds. These radiations have fundamentally modified the biosphere, ultimately affecting the ecology and evolution of multicellular and unicellular organisms alike.

Our focus here is on the first steps in the transition to multicellular life: the evolution of multicellularity per se, and the early evolution of nascent multicellular populations. A necessary early step in the transition from unicellular to multicellular life is the evolution of a mechanism that keeps cells together [1,2,3]. In some cases, the population may have already expressed a multicellular phenotype plastically, as is known to happen in some algae [4,5] and choanoflagellates [6], and the transition in question is from facultative to obligate multicellularity. In other cases, the multicellular phenotype may be entirely novel, resulting in a transition from a purely unicellular population to an obligately multicellular one. Both processes have been observed in microbial evolution experiments in which a selective pressure was applied to unicellular laboratory populations [7,8,9,10].

The resulting populations consist of clusters of clonally related cells. These clusters have traits that did not exist in the unicellular population, for example the cluster diameter, cluster shape, and the number of cells per cluster. For selection to act on emergent multicellular traits, however, they must be heritable.

Understanding the emergence of heritability at the multicellular (or, more generally, collective) level has been treated as one of the central problems of the Major Transitions framework by Maynard Smith and Szathmáry [11]. For example, Michod and Roze (2001) [12] wrote that “The basic problem in creating new evolutionary individuals involves generating heritable variation in fitness at the group level”. Griesemer (2001) [13] described the implementation of heritability at new levels of organization as part of “the problem of evolutionary transition”. Okasha (2006) [14] asked “Does the heritability of a collective character depend somehow on the heritability of particle characters?” and called this question “crucial”.

In some cases, the perceived problem is based on an assumption that substantial collective-level heritability appears only late in a Major Transition. Rainey and Kerr (2010) [15], for example, consider such a transition complete “… when the higher-level entities become Darwinian individuals, that is, when populations of these organisms display variation, heritability, and reproduction”. Trestman (2013) [16] describes an incipient multicellular organism as “… a unit of behavioral organization… before there was any aggregate-level development or inheritance”. Simpson (2011) [17] treats the presumed low heritability at the collective level as a factor making Major Transitions “impossible” to understand through “standard theory”.

Several authors have gone further, claiming that the emergence of substantial heritability at the multicellular (or collective) level requires some evolutionary process subsequent to the formation of collectives. Müller and Newman (1999) [18] refer to the “pre-multicellular and early multicellular world” as “pre-Mendelian” and, in the case of metazoans, conclude that the establishment of the major body plans was required for the “… increasingly unique matching between genotype and phenotype [that] led ultimately to Mendelian heritability” [19]. Alvarado and Kang (2005) [20] consider the emergence of stem cells as “… a fundamental evolutionary adaptation that allows multicellular organisms to satisfy Darwin’s conditions of heritability and variation in fitness”. The clearest expression of this view is probably that of Libby and Rainey (2013) [21]: “It is not sufficient to assume that Darwinian properties inherent in the lower level units are simply ’moved up’ to the higher level. Variation, heritability and reproduction are derived properties and their emergence at the group level requires an evolutionary explanation. Somehow, during the transition to multicellularity, groups evolved Darwinian characteristics. Just how these emerged is a problem of seminal significance”.

Richard Michod has expressed this view in a series of papers beginning in the late 1990s and continuing to the present. For example, Michod [22] claims that the transition to multicellularity requires that “… ways must have been found to ensure the heritability of the properties of these ensembles of cells”. Michod and colleagues consider the de-coupling of survival and reproduction (accomplished by the evolution of a sterile soma) a requirement for the heritability of multicellular traits [23,24,25]. This is sometimes expressed as “the reorganization of the basic components of fitness” [24], and Michod considers his influential two-locus modifier model [12] a solution to the problem. Most recently, in response to Rainey and DeMonte’s (2014) ([26]) characterization of collective-level heritability as a “derived state… requir[ing] evolutionary explanation,” Michod wrote, “I agree that collective level heritability is a derived state, a derived state that is explained by the modifier models considered here” (2022) [27].

Here, we consider the heritability of collective-level traits in a laboratory-derived multicellular organism. By working with strains that are isogenic except for the mutations we introduced, we ensure that no evolutionary change subsequent to the formation of multicellular groups has occurred.

In a previous analytical model, we showed that, given reasonable assumptions, the heritability of a collective-level trait is not only substantial but higher than that of the cell-level trait of which it is a function under a wide range of conditions [28]. Our approach relied on the fact that all collective-level traits can, in principle, be expressed as functions of cell-level traits. Such functions could, in principle, be very simple. Imagine, for example, a simple clump of cells with no extracellular material: the collective-level trait of cluster volume would be a linear function (the sum) of the cell-level trait of the cell volume (assuming a fixed cell number). At the other end of the spectrum, the function could be so complex as to be intractable. It may, for example, include cell–cell communication, complex feedbacks, and interactions with the environment, such that the cell-level phenotype is only one of several arguments.

Because it is a better fit to our skill set, we focused our analytical model on the simple end of the spectrum: cases in which the collective-level trait is a linear function of the cell-level trait. In a series of simulations, we extended the model to collective-level traits that are increasingly complex functions of cell-level traits, including some directly tied to fitness [28]. These simulations showed that the central result of the analytical model holds for nonlinear functions as well. In every case, the heritability of the collective-level trait is not only substantial but exceeds that of the cell-level trait under a wide range of conditions.

In this paper, we show that the assumptions and predictions of our analytical model [28] hold in a real-world engineered biological system. Our focus is on an emergent cluster-level trait whose value is a function of a single cell-level trait: cluster size at division. In experimentally-evolved multicellular ‘snowflake’ yeast, the size that clusters can grow to before reproducing (see Appendix A) is a simple linear function of the aspect ratio of the component cells [29,30,31]. By using strains that differ only by a single mutation, we eliminate the possibility that these results depend on any evolutionary process that occurs subsequent to the initial formation of multicellular collectives.

Through a combination of analytical modeling, experimental manipulation, and simulations, we show that substantial heritability of a collective-level trait exists from the first step in the evolution of multicellularity. Heritability (in this paper we use broad-sense heritability, as we are considering asexually reproducing organisms [32]) of cluster size at reproduction in fact exceeds that of the underlying cell-level trait, the cellular aspect ratio, across a wide range of conditions. A key insight behind our result is that clonal multicellular groups have the ability to average out stochastic variation in the cellular phenotype, which reduces the within-genotype variance at the group level. This means that nongenetic variation is a smaller proportion of the total phenotypic variation at the group level than at the cell level, resulting in higher heritability. To illustrate this point, consider a simple example of a multicellular trait that is a linear function of a cell-level trait: cluster volume. The cluster volume is simply the sum of the cell volumes in a cluster, assuming a fixed cell number. If the cell volume varies randomly among cells within a genotype due to environmental or developmental noise, then the cluster volume will also vary among clusters within a genotype. However, the variance in the cluster volume will be smaller than the variance in the cell volume, because the random fluctuations in the cell volume will tend to cancel out when summed over many cells. Therefore, the heritability of the cluster volume will be higher than the heritability of the cell volume. These results show that novel multicellular traits can be remarkably heritable and thus subject to adaptive change, at the very origin of a multicellular life cycle.

## 2. Results

### 2.1. Experimental System

We experimentally examined the relative heritability of both a cellular trait (the cellular aspect ratio) and an emergent multicellular trait (the number of cells in the cluster at reproduction) using the snowflake yeast model system [7]. Prior biophysical modeling (which we confirmed experimentally, see Figure 1) suggests that there is a linear relationship between the size to which clusters grow before they reproduce and the cellular aspect ratio [33]. Snowflake yeast grow as branched tree-like groups of cells. When their cells are more elongate, this reduces cellular crowding within the interior of the cluster, slowing the rate of stress accumulation that ultimately causes the cluster to fracture. As a result, genotypes with a higher cellular aspect ratio are capable of forming larger clusters [31,33].

We deleted three cell cycle regulatory genes (*AKR1*, *ARP8*, and *CLB2*), generating a range of mutants that varied in their mean cellular aspect ratio. In each case (and in the wildtype), we generated otherwise isogenic unicellular and multicellular versions of these strains by leaving *ACE2* functional or deleting the *ACE2* open reading frame, respectively [34] (Figure 1A).

In both unicellular and multicellular isolates, we measured the cellular aspect ratio (ratio of length to width) of cells that were a single division old. Because the cellular aspect ratio varies with replicative age in yeast [35], this provides an upper bound estimate of the heritability of this trait at the cell-level by excluding age-dependent phenotypic variation. *arp8*Δ, *akr1*Δ, and *clb2*Δ increased the mean aspect ratio from 1.26 in the ancestor to 1.41, 1.49, and 2.02, respectively (Figure 1B; *F*${}_{3399}$ = 642.7, *p* < 0.0001, ANOVA, all means significantly different at *p* = 0.001 with Tukey’s HSD test). Whether or not Ace2p was active did not significantly affect the cellular aspect ratio in any of our genotypes (Figure 1B, t=−0.96, p=0.33 for the wildtype, t=1.12, p=0.26 for *arp8*Δ, t=−0.69, p=0.49 for *akr1*Δ, t=−0.29, p=0.76 for *clb2*Δ; two-tailed *t*-tests).

To measure the number of cells in the cluster at reproduction (*N*), we measured the cross-sectional area of the cluster just before division using time-lapse microscopy. For each genotype, we measured the mean cellular volume and packing fraction and then used this data to infer the number of cells in the cluster at fracture (Figure 1C). We validated this approach using volumetric electron microscopy (Appendix A). The cluster size at reproduction is a key life history trait for snowflake yeast, setting an upper bound on group size, which strongly affects the fitness. The cluster size at reproduction in these strains is an approximately linear function of cellular aspect ratio (y=2549.8x−2526.6, $r^{2}=0.96$, Figure 1D). To more comprehensively examine the functional form of this relationship (our experiments were limited to four genotypes), we adapted the 3D biophysical simulation from [33], which allowed us to simulate snowflake yeast growth until the packing strain arising from the cellular division caused fracture (Figure 1E). Simulating the growth and fracture of 25 clusters for cellular aspect ratios ranging from 1 to 2.2 (in increments of 0.1), we confirmed that the group size at reproduction indeed was an approximately linear function of the cellular aspect ratio (y=1240x−485, $r^{2}=0.99$).

### 2.2. Quantifying Heritability

The breeder’s equation of quantitative genetics shows that a trait’s response to selection (*R*) is the product of the selective coefficient (*S*) and the heritability of the trait under selection (*h*^2^): $R=h^{2}S$. The suitable measure of heritability in the breeder’s equation is the realized heritability of the trait, which is typically the narrow-sense heritability in sexually-reproducing organisms or the broad-sense heritability (*H*^2^) in asexual populations. This formulation predicts the response to a given selection differential [32]. For simplicity, we consider asexual populations. The response to selection in an asexual population, then, is $R=H^{2}S$ [32].

Thus, in principle, we could predict the response of a nascent multicellular population to a given strength of selection if we could predict *H*^2^. Doing so is not trivial, though. Heritability is not an inherent property of organisms or species. Rather, it is a population-level statistical measure that applies only to a particular trait in a particular population at a particular time and is highly sensitive to the population composition [36].

We used a variance partitioning approach [32] to calculate the broad-sense heritability of the cellular aspect ratio and group size at division from our experimental data (see the Appendix A for a formal derivation). Because each cell cycle mutant has a different genetic mean cellular aspect ratio, the heritability of both the cell and group-level traits will depend on the frequency of each genotype in the population. Therefore, we measured the heritability of both the cellular aspect ratio and cluster size at reproduction by simulating all possible hypothetical populations, generating populations of 1000 individuals by subsampling, without replacement, from our experimental data (Figure 2). Because it is difficult to visualize four-dimensional space, which is requred if we simultaneously vary the frequency of four genotypes, we examined all possible three-member populations, varying the frequency of each genotype from 2.5 to 95%. Setting a maximum genotype frequency of 95% ensures that the population never becomes monomorphic, a pathological case where heritability is undefined. In nearly all possible populations, the multicellular trait, the number of cells in the group at reproduction, is more heritable than the underlying cell-level trait, the cellular aspect ratio, despite the fact that the cellular aspect ratio is the only trait being modified directly via mutation (Figure 2).

## 3. Discussion

During a major evolutionary transition, organisms form novel collectives that become more complex and biologically individuated by gaining adaptations [37]. For adaptations to occur at the collective level, nascent collectives must not only be capable of reproduction but must also possess heritable variation in traits that affect fitness [38]. In this paper, we focus on the origin of collective-level heritability during the transition to clonal multicellular life. We show that contrary to previous suggestions e.g., [12,15,16,17,18,19,21,22,23,24,25,26,27], substantial collective-level heritability exists as soon as clonal multicellular groups form, requiring no subsequent evolutionary change.

Instead, we confirm predictions by a previous analytical model [28], that the heritability of the multicellular trait can in fact be higher than that of the underlying cellular trait, even when the possibility of evolutionary change subsequent to the origin of multicellularity is excluded. Although the ratio of cellular and multicellular heritabilities depends on the proportion of each genotype in the population, it is in nearly all cases greater than one.

Clonal development, the “staying together” [2] of the products of mitosis, is not the only path to multicellularity. Aggregative multicellularity, the “coming together” [2] of previously free-living cells to form a multicellular structure, is common in a wide variety of prokaryotic and eukaryotic groups, for example the cellular slime molds and myxobacteria, and this mode has evolved independently in six of the eight major groups of eukaryotes [39,40].

Our focus, though, is on clonal multicellularity. Because the genetic structure of groups is fundamentally different between clonal and aggregative multicellularity, the implications of this work apply only within this scope. It is a rather large scope, though, as clonal multicellularity is found in all three domains of life and has evolved many times independently. The frequently-cited estimate of 25 independent origins [41] is certainly an underestimate, possibly by two-fold or more [42,43,44].

Prior work has examined group-level heritability mainly via mathematical modeling. Bourrat [45] has shown that, under idealized conditions, the heritability of the group and its constituent cells can be identical. Specifically, when group-level traits are an additive function of cell-level traits, and the cells of a given genotype are identical (i.e., no non-heritable phenotypic heterogeneity), then group-level traits arise as a direct consequence of the heritability of their constituent cells. These assumptions will often be violated in natural populations, however, as non-genetic contributions to both cellular and group-level phenotypes are ubiquitous (see below for further discussion), and many functions relating cell to group-level traits are non-additive.

In a group selection model, Queller included terms for within-group and among-group heritabilities in his application of the Price equation [46] framework to a group selection scenario [47]. Queller’s model is fundamentally different from and incommensurable with ours, most importantly because it is focused on a single individual-level trait subject to within-group and among-group selection, while ours is focused on a group-level trait (cluster size) that is distinct from the individual-level trait of which it is a function (cell shape). Queller’s and other approaches based on the Price equation e.g., [48,49] have the advantage that they allow a response to selection to be decomposed into a within-group component and an among-group component, which our approach cannot. However, Queller’s and related approaches do not allow one to calculate individual-level heritability, group-level heritability, or the relationship between them without knowing the strength of and response to selection; that is, they only allow estimation of the heritabilities in retrospect, after selection has occurred. However, since the nature of heritability as a population-level parameter guarantees that its value changes each time selection is applied (outside of pathological cases), values of heritability estimated in retrospect (after selection) cannot be used to predict future responses to selection. In contrast, our model neither considers nor depends on selection. Rather, it allows estimation of the heritability and the prediction of future responses to selection from variables that can be measured before selection or even in its absence.

While it might seem surprising that an emergent multicellular trait can be more heritable than a cell-level trait directly encoded for by genes, clonal multicellular groups possess a powerful advantage over isolated cells: the ability to average out stochastic variation in the cellular phenotype [28]. Because of this averaging effect, cluster phenotypes are less variable within a genotype than cell phenotypes, so non-genetic variation is a smaller proportion of total phenotypic variation ([28], see also the derivation in the Appendix A).

Although it is routinely estimated and widely recognized as useful in predicting responses to selection in clonally reproducing multicellular organisms [50,51,52,53,54,55,56,57,58], *H*^2^ is typically assumed to be 1 in asexually reproducing microbes [59,60,61,62,63,64]. This assumption is so widespread that the concept of heritability is rarely addressed in discussions of microbial evolution; rather, the response to selection is simply assumed equal to the strength of selection (R=S). However, *H*^2^ is never 1 for a continuously varying trait in a real population. Because broad-sense heritability is defined as the ratio of genetic variance to total phenotypic variance, which includes environmental variance, the only way for *H*^2^ to equal 1 is for there to be zero non-genetic variance.

In reality, phenotypic heterogeneity is widespread even among genetically identical individuals reared in carefully controlled environments. This phenomenon is well documented among microbes, but it is also true for diverse plants, fungi, and animals [65] and for traits as diverse as gene expression [66,67], antibiotic [68] and formaldehyde [69] resistance, motility [70], phototaxis [71], cell number [72], predator evasion behavior [73], and number of sense organs [74].

While the emergence of multicellular heritability is widely regarded as a significant challenge in this major evolutionary transition e.g., [1,12,15,16,17,18,19,21,22,23,24,25,26,27,75,76,77], we have shown that substantial multicellular heritability may in fact arise spontaneously, as a side-effect of group formation. Our experimental design excludes the possibility that this results from evolutionary change subsequent to group formation, since the unicellular strains are genetically identical to the multicellular strains, aside from the *ace2* deletion.

If it were true that the heritability of multicellular traits only arose as the outcome of some adaptive process subsequent to the origin of multicellularity, this would, as previous authors have pointed out [14,15,16,78,79,80,81], engender a chicken-and-egg problem for the evolution of complex multicellularity: how could heritability arise as an adaptation when heritability is a prerequisite for adaptation? We have shown analytically and empirically that no such problem exists. Multicellular traits are heritable as soon as they come into existence, and this is a straightforward result of the way variation is partitioned. Adaptive modification of multicellular-level traits is thus bootstrapped into feasibility, and subsequent adaptations can build upon earlier ones. We speculate, and indeed see no reason to doubt, that the spectacular complexity that characterizes a few multicellular clades arose through a sequence of adaptive changes that started with a process similar to the one we have described.

## 4. Methods

### 4.1. Strain Construction

All yeast strains were produced from a Y55 strain background made homozygous at all loci via sporulation and selfing, as previously used by the Ratcliff laboratory [34]. All strains were based on the GOB8 strain bearing a homozygous deletion of the *ACE2* open reading frame with the KanMX cassette, resulting in an *ace2*Δ:KanMX/*ace2*Δ:KanMX, producing small constitutively multicellular snowflake yeast. Cell-lengthening mutations were identified from previous work showing that disruption of *CLB2* results in an alteration of filamentous growth [82] and screens of genes known to alter the aspect ratio or filamentous growth form of yeast cells [83].

Strains (Appendix A) were produced via standard yeast genetics protocols. Cells were transformed via the lithium acetate + polyethylene glycol transformation method [84]. All the deleted genes were removed via transformation using PCR products of plasmid pYM25 [85], creating ends-out gene deletion cassettes bearing the hygromycin-resistance hphNTI gene with 50 base pairs of flanking sequence just outside of each reading frame. The sequences can be seen in Appendix A. Upon selection of the heterozygous deletion strains on Yeast Peptone Dextrose (YPD) + hygromycin plates, colonies heterozygous for the deletion were sporulated, the tetrads were dissected onto YPD + hygromycin plates, and the spores were allowed to self-mate before single-colony purification for analysis.

### 4.2. Cell Culture

The yeast were cultured in Yeast Extract Peptone Dextrose (YEPD) media (1% yeast extract, 2% peptone, and 2% dextrose) at 30 °C. In large-diameter tubes (25 mm), each strain was cultured in 10 mL of YEPD with rapid mixing (250 rpm).

### 4.3. Measuring the Cellular Packing Fraction

To measure the packing fraction of each genotype, we took 1 mL of each cell culture and stained them with DAPI using the following protocol: first, we transferred 500 μL of each cell culture to a 1.5 mL microcentrifuge tube and replaced the YEPD media with 500 μL of 70% ethanol. Then, we shook the tubes at 1300 rpm for 5 min at 25 °C and removed the ethanol by washing, pelleting, and resuspending in 1% phosphate buffered saline solution. We added 0.1% DAPI stock solution (1.25 mg/500 mL) and vortexed vigorously. We incubated the cells for 5 min at 25 °C, then diluted the stained cells 1:10, and imaged the individual clusters without lateral compression on a Nikon Eclipse Ti under brightfield illumination using a 10x objective. From this, we calculated the effective radius of each cluster as:(1)$$\begin{array}{c} r_{eff}=\sqrt{\frac{Area}{\pi }}. \end{array}$$

We approximated the volume of each cluster by calculating the volume of a sphere with $R=r_{eff}$. We then placed a coverslip over the cluster and applied pressure until it was a cellular monolayer, allowing us to image the DAPI-stained nuclei using a 20x objective. This allowed us to count the number of cells per cluster ($N_{cell}$) using ImageJ-FIJI. We approximated the volume of 100 cells ($V_{cell}$) for each genotype (ensuring each was just one cellular generation old) by calculating the volume of a prolate ellipsoid so that its minor and major axes were equal to the short and long axes of a cell, respectively. We used this information and the formula:(2)$$\begin{array}{c} \phi =\frac{N_{cell}*V_{cell}}{V_{cluster}}. \end{array}$$

To validate the method used to estimate the cell number via time-lapse microscopy, we used previously published data of 20 clusters of snowflake yeast (Δace*2*) sectioned using SBF-SEM with 50 nm z sections [29] and used these spatial coordinates to estimate a perceived cross-sectional area of the cluster that would be similar to a measurement obtained from optical microscopy. Appendix A shows that this method accurately estimated the number of cells within the cluster ($R^{2}=0.903$), given knowledge about the 2D cross-sectional area, cell size, and cell packing fraction.

### 4.4. Quantifying Cellular Aspect Ratio

To measure the cellular aspect ratio, we grew samples both with and without functional *ACE2* alleles (i.e., uni- and multicellular versions of these genotypes) overnight at 30 °C in YEPD, shaking at 250 RPM. After that, we grew these samples for 2 days until they reached stationary phase. Next, we used calcofluor-white to stain the cell walls, incubating them in the dark for 30 min in a one micromolar solution, and imaged them with a Nikon Eclipse Ti microscope under ultraviolet excitation for blue fluorescence using a 20x objective. We measured the aspect ratio of 50 individual cells for the ancestor and each cell cycle mutant, repeating the experiment both with and without functional *ACE2* strains. Cell boundaries were identified manually using ImageJ-FIJI. In all cases, we only considered cells with a single bud scar, mitigating issues stemming from possible age-dependent variation in cellular aspect ratio.

### 4.5. Quantifying Cluster Size at Division

We determined the size of clusters at division using time-lapse brightfield microscopy. The clusters were inoculated into 1 μL drops of YEPD medium, grown overnight, and imaged every five minutes at using a 4x objective. Using ImageJ-FIJI, we then measured the radius of 50 clusters per each genotype in the timestep just before multicellular division. Using the formula described above, we calculated the volume of each cluster, and knowing the mean cell size and packing fraction of these genotypes, we were able to estimate the number of cells in each group just prior to division.

### 4.6. Calculating Heritability by Partitioning Variance

There are two common methods to estimate broad-sense heritability: the regression approach and the variance approach. These methods are not equivalent, and they may yield different estimates of heritability. There is some debate in the field about which method is superior [86,87]. In this paper, we used both approaches to estimate broad-sense heritability, leveraging the favorable features of each, though we acknowledge that others may have chosen a different approach. We used the variance approach in our formal model (Appendix A) to examine how different sources of phenotypic variation affect the heritability of cell and group-level traits and under what conditions the group-level trait is more heritable than the cell-level trait. In our experimental system, we used the regression approach to estimate the heritability of both traits from our empirical data. This approach allowed us to measure how genetic variation among genotypes translates into phenotypic variation among cells and groups and how this affects their response to selection. Below, we describe how we implemented the regression approach.

For each population simulated in Figure 2, we calculated the broad-sense heritability as the correlation coefficient between the parental cell phenotype α (cellular aspect ratio) and the offspring cell phenotype $\alpha^{\prime }$ for cellular aspect ratio, partitioning the phenotype into its key components [36]:(3)$$\begin{array}{c} H^{2}\equiv Corr\left(\alpha ,\alpha^{\prime }\right)=\frac{E[\left(\alpha -\bar{\alpha }\right)\left(\alpha^{\prime }-\bar{\alpha^{\prime }}\right)]}{Var(\alpha )}, \end{array}$$
where
(4)$$\begin{array}{c} \alpha =G+E+S\;\alpha^{\prime }=G+E^{\prime }+S^{\prime }. \end{array}$$

In this equation *G* is the genetic mean of the aspect ratio, *E* represents the environmental effects with the mean $\bar{E}$ and variance $\sigma_{E}^{2}$, and *S* is the intrinsic variation in the expression of the trait, which we refer to as developmental noise. Here, we assume that *S* is white noise, with mean zero and variance ($\sigma_{S}^{2}$).

### 4.7. Biophysical Simulations

We simulated snowflake yeast by adapting the spatially explicit approach described in [31,33]. In these simulations, cells were modeled as prolate ellipsoids. Briefly, new cells were procedurally added to the surface of existing cells in the cluster, resulting in cluster growth in a manner analogous to the way snowflake yeast grow. Each cell was allowed to divide as long as there was space for an offspring cell. All elements of the simulation were held constant, as shown in Figure 2D, varying only the input distributions of the cellular aspect ratio. See the following section for a more detailed description of the simulation.

To generate snowflakes, new cells were added in an analogous manner to cellular growth in actual snowflake yeast. Every generation, each cell in the cluster attempted to divide. They first budded from their distal pole (i.e., θ=0±10 degrees), with subsequent cells budding at a polar angle θ=45 degree, and randomly chose an azimuthal angle from a uniform distribution ϕ∈[0,2π]; in other words, after the first bud, the cells generally appeared along a “budding ring” at the distal pole of the cell. There was a 20% chance that the first bud would appear along this budding ring instead of exactly at the pole. After 3 bud scars, there was a 50% chance that new cells would bud on the proximal side (π−θ) instead of the distal side. If new bud scar locations were located within 1.17 microns of a previous bud scar on the cell surface, then the new bud was unable to form, and the mother cell lost its chance to generate a new cell for that generation. The orientation of the new cell was determined by the surface normal to the mother cell at the position of the bud site. The cellular aspect ratio of each new bud was drawn from a normal distribution centered on the mean value α with standard deviation α/10.

After each generation of cell division, there was a chance that two cells would overlap due to the proximity of their bud scars or because two cells grew into the same space, etc. The amount by which any two cells overlapped was computed by d=D−R if d<0, where *d* is the overlapping distance, *D* is the distance between the centers of two cells, and $R=R_{1}+R_{2}$ is the sum of the two cell’s equatorial radii. We modeled the energy associated with this overlap as Hertzian (i.e., $u∼d^{5/2}$). The total energy of the overlaps was then $U=\sum_{i}u$ where *i* ran over all pairs of cells. The total overlapping energy was recorded after each generation of cell division, and if there were more than 20 budding events, then it was after every 20 budding events. Once the total overlapping energy succeeded a threshold amount $U_{overlap}$, the simulation was stopped. This was the final size of the cluster. In total, 25 groups were simulated in this manner for each aspect ratio.

## Figures and Tables

**Figure 1 genes-14-01635-f001:**
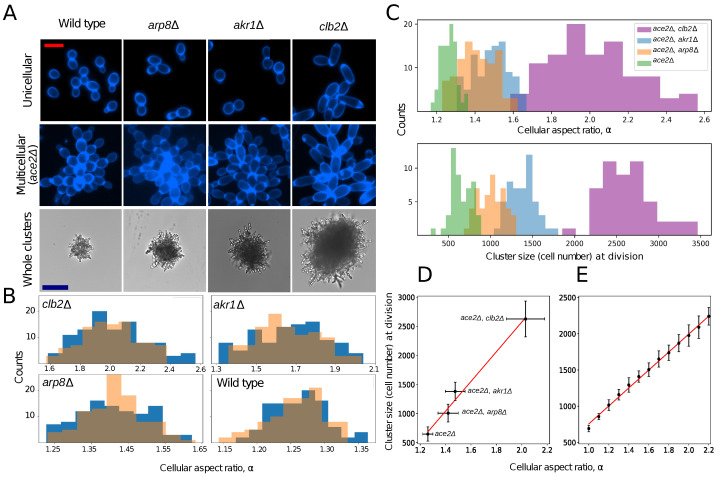
Experimentally modifying the cellular aspect ratio. (**A**) Cell cycle mutants with functional (top row) or nonfunctional (second row) *ACE2* alleles. The bottom row shows representative brightfield images of multicellular *ace2*Δ, demonstrating how cellular elongation increases the cluster size. The scale bar in the top two rows of A is 10 μm and in the bottom row of A is 50 μm. (**B**) Distribution of the cell aspect ratio for each genotype with functional *ACE2* (orange bars) and nonfunctional *ace2* (blue bars). (**C**) Distribution of the cellular aspect ratio (top) and number of cells at fracture (bottom) for the four multicellular genotypes in our experiment. The cluster size at division is an approximately linear function of the cellular aspect ratio. Empirical data in (**D**) (y=2549.8x−2526.6, $r^{2}=0.96$) and the results from a biophysical simulation of the cluster growth and fracture in (**E**) (y=1240x−485, $r^{2}=0.99$).

**Figure 2 genes-14-01635-f002:**
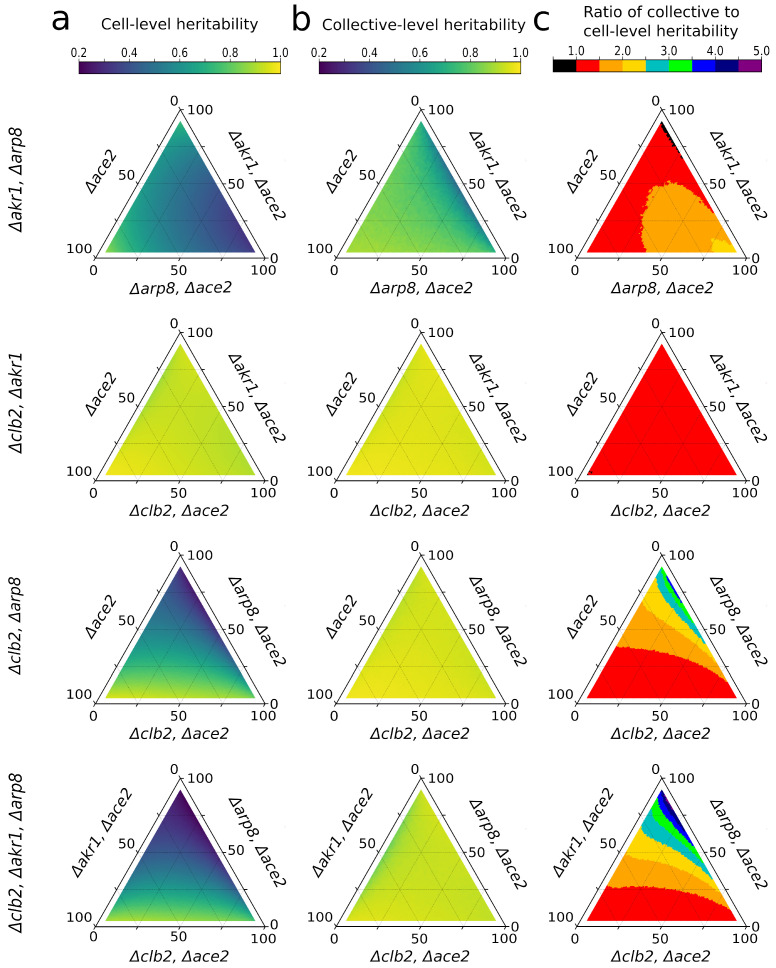
The heritability of an emergent multicellular trait exceeds that of the corresponding cell-level trait over a wide range of simulated populations. Each triangle plot represents the heritability of the cell-level trait, the cellular aspect ratio (**a**), the collective-level trait, the cluster size at reproduction (**b**), or the ratio of the collective to cell-level trait heritabilities (**c**). Shown are all possible 3-way combinations of the four genotypes (each row being a different population composition), across all possible population frequencies ranging from 2.5 to 95% per genotype.

## Data Availability

All data and code required for analyses is available in the Appendix A.

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
