# Peer review of "Spontaneous Emergence of Multicellular Heritability"

_genes, 2023, doi:10.3390/genes14081635_

Round 1
Reviewer 1 Report
As a philosopher of biology with a background in evolutionary biology, I will focus my report on the conceptual aspects of the paper rather than the experimental ones. The paper addresses the topic of heritability in major transitions, which has been less studied than fitness. The paper is interesting and deserves to be published, as it makes some progress on this topic. The authors show that there is a linear relationship between the cluster size of a collective and the cellular aspect ratio, and that based on a previously published analytical model, the heritability of cluster size is higher than that of cellular aspect ratio.
Major comments: I recommend that the authors revise their paper to clarify some of their claims:
- First, the claim that collective-level heritability can be higher than individual-level heritability might seem surprising and intuitively wrong to an average reader. I understand that this claim relies on the analysis of a previous paper and that the current paper provides supplementary material with derivations. However, heritability is a statistical summary that can be tricky to interpret, so it is important to convince the readers and explain this claim more thoroughly.
- Second, I am puzzled by the developmental noise term at the cluster level from a causal perspective. If this term does not originate from the environment, it must come from the genetic makeup or some other heritable factor. In that case, I wonder why it would not be accounted for at the cell level in a cluster. Some clarification on this point would be helpful and appreciated.
- Third, (and this might reflect my philosophical background), it would be good to have a clear picture of the causal relationship between the particle trait and the collective trait. I understand that there is a linear relationship between them, but at the same time, there must be some source of stochasticity. If this source is environmental, then it should be included in the environmental term. If it is cellular but not genetic, then it would suggest that the cell aspect ratio is a major determinant of the collective trait but not sufficient. The fact that the mean is zero is compatible with this explanation. Similarly, if it is cellular and genetic, but mediated by indirect genetic effects, it could also be compatible with your analytical model.
Minor comments:
- The introduction does not define heritability, which is problematic since there are different definitions of this concept. The authors should define what they mean by heritability early on, before using it in their arguments. The authors do define it later, but I think it would still be valuable to have it at the beginning.
- Related to the previous point: There are at least two notions of heritability used in evolutionary theory: the regression approach and the variance approach. There is some debate about which one is an estimate of the other, and whether one is superior to the other (see Okasha 2010, Mamelly 2004, Bourrat 2022). In your manuscript, you seem to use both approaches, but you favor a breeder’s equation notion (regression) of heritability when measuring it rather than defining it, and a variance approach when defining it. It would be good to clarify which notion of heritability you endorse and why, and acknowledge that your choice might be disputed by others.
· Related to the previous point: on Page 7 you write “Heritability in this formulation is the narrow-sense heritability, the proportion of phenotypic variation explained by additive genetic variation, and this formulation predicts the response to a given strength of selection in an obligately sexual population (Lynch et al. 1998).” I do not think this is correct. First, the heritability of the breeder’s equation does not require an obligately sexual population. Second, I don’t think it refers to narrow sense heritability. If there is gene/environment covariance, heritability might be greater than 1 for instance. So to obtain narrow sense heritability, one needs to make further assumptions.
- Page 7, the following sentence: “For simplicity, we consider asexual populations…” seems to contradict the previous one.
- On pages 9-10, I was surprised to read that the approaches based on the Price equation are different from yours. The breeder’s equation can be derived from the Price equation (see for instance Walsh & Lynch 2019).
- Pages 9-10, same paragraph (l. 216-234), it is unclear to me why your approach cannot be decomposed into components. Could you explain why?
- Line 97: “could be will be so” typo. -> “could be so”
- Line 251: ‘which includes genetic variance’ should it be “environmental” -> “which includes environmental variance”
- I have argued (https://doi.org/10.1007/s12064-019-00294-2) in a publication that heritability at the collective level is equal to the heritability at the particle level (in the absence of developmental noise and mutations), contrary to what is classically claimed for additive collective traits. So I agree with you that collective-level heritability should not be necessarily viewed as being lower than particle-level heritability, but for me, once one has the same level of information for both the particle and collective level description, the two should be equal. Do you disagree or agree with this claim? Supplementary material:
- In the supplementary material, N becomes n at some point between equations 4 and 5 (also the definition of Var(N) has an additional closing bracket. -> “In the supplementary material, n becomes N at some point between equations 4 and 5 (also the definition of Var(N) has an extra closing bracket.”
- The definitions of parent and offspring should be given for cells and collectives. Are they parent and offspring populations? If they are measured on the same timescale at both levels, then I would use parent and ‘descendant’ to include more remote generations from the parent. (the terms with primes should also be defined) -> “The definitions of parent and offspring should be given for cells and collectives. Are they parent and offspring individuals or populations? If they are measured on the same timescale at both levels, then I would use parent and ‘descendant’ to include more remote generations from the parent. (the terms with primes should also be defined as either individuals or populations)”
- From equations 4 to 5, please explain why you get this formula (equation 5). In particular, why do you divide by n (should be N I think)? -> “From equations 4 to 5, please explain how you derive this formula (equation 5). In particular, why do you divide by N?”
Signed,
Pierrick Bourrat
Reviewer 2 Report
heritable traits at the multicellular level during the transition from unicellular to multicellular life. The authors present an analytical model and experimental evidence using a laboratory-derived multicellular organism, specifically a "snowflake" yeast. The results indicate that substantial heritability of collective-level traits exists from the very beginning of multicellularity's evolution and, in many cases, surpasses the heritability of underlying cell-level traits. This finding challenges previous assumptions that heritability at the multicellular level requires subsequent evolutionary changes. The objectives of this manuscript are clear, and the research is of significant interest. While I don't have major comments, I do have a few suggestions below:
The introduction provides a comprehensive overview of the study, but it could benefit from being more concise and focused. Consider highlighting the main research question, the study's significance, and the specific aims of the investigation in a more succinct manner.
The "Results" section is quite extensive, and it might be helpful to divide it into subsections for better readability. Grouping similar experiments and outcomes under separate headings can help readers navigate the content more easily.
In the abstract, the manuscript mentions the use of analytical models, synthetic biology, and biologically-informed simulations. However, it is not entirely clear how these methods were applied in the research. Providing a more detailed explanation of each experiment and how they relate to the main research question would enhance the clarity of the manuscript.
Consider including figures or diagrams to illustrate the experimental setup, analytical models, and simulation results. Visual representations often aid in understanding complex concepts and can enhance the overall readability of the paper.
Some sentences in the manuscript are quite lengthy and complex, making it challenging for non-native English readers to understand them. Aim for clear and concise sentence structures to improve readability.
Round 2
Reviewer 1 Report
The authors have responded adequately to my comments with explanations that clarify their points.
I do have a further comment about the explanation of the shift in values of heritability from lower-level heritability to higher-level heritability. This comment might need to be addressed in the manuscript, but I leave this to the authors' decision. I am mostly writing this comment because I am very interested in the topic.
The authors rightly point out that cluster volume variance is lower than cell volume variance assuming the same population. They claim that this is the reason why collective-level heritability is higher than cell-level heritability. I agree. However, I would argue that the reason is that what is considered non-genetic variance at the cell level has been ‘integrated’ into the collective-level trait measurement when computing heritability at the cluster level. I am unsure, as a result, that talking about the heritability of a cell trait vs a cluster trait simpliciter is the most accurate description of what the authors measure. I think that the authors’ comparison might be more accurately viewed as a comparison between the heritability of cell volume within a cluster vs the heritability of cell volume between clusters, where the former would be lower than the latter. The reason why the former would be higher is precisely because there is less variance in volume between clusters volume than within a cluster. Note that this mirrors the within vs between-group selection, and for that reason I think it might be a good idea to mention it. Also, within and between group selection has been interpreted as lower and higher level selection by many people, but it should be noted that lower/higher and within/between are in many respect quite different, hence why I suggest mentioning this point.